# Transient domains of ordered water induced by divalent ions lead to lipid membrane curvature fluctuations

O.B. Tarun[1], H.I. Okur[1,2], P. Rangamani [3] & S. Roke[1]*

Cell membranes are composed of a hydrated lipid bilayer that is molecularly complex and diverse, and the link between molecular hydration structure and membrane macroscopic properties is not well understood, due to a lack of technology that can probe and relate molecular level hydration information to micro- and macroscopic properties. Here, we demonstrate a direct link between lipid hydration structure and macroscopic dynamic curvature fluctuations. Using high-throughput wide-field second harmonic (SH) microscopy, we observe the formation of transient domains of ordered water at the interface of freestanding lipid membranes. These domains are induced by the binding of divalent ions and their structure is ion specific. Using nonlinear optical theory, we convert the spatiotemporal SH intensity into maps of membrane potential, surface charge density, and binding free energy. Using an electromechanical theory of membrane bending, we show that transient electric field gradients across the membrane induce spatiotemporal membrane curvature fluctuations.

[1] Laboratory for fundamental BioPhotonics (LBP), Institute of Bioengineering (IBI), and Institute of Materials Science (IMX), School of Engineering (STI), and Lausanne Centre for Ultrafast Science (LACUS), École Polytechnique Fédérale de Lausanne (EPFL), CH-1015 Lausanne, Switzerland. [2] Department of Chemistry and National Nanotechnology Research Center (UNAM), Bilkent University, 06800 Ankara, Turkey. [3] Department of Mechanical and Aerospace Engineering, University of California, San Diego, La Jolla, CA 92093, USA. *email: sylvie.roke@epfl.ch

Cell membranes are in a constant state of structural flux[1,2]. Lipid bilayer membranes provide a rich environment for the formation of diverse structures and environments. Membrane structuring is commonly described in terms of lipid segregation and measured by probing the hydrophobic core of a membrane[3–5], or tagged lipids[6–10]. While this provides information about lipid–lipid interactions, tagging may also modify the membrane. Both methods also largely ignore the role of water, electrostatic and hydrogen bonding interactions, and the influence of the electric double layer. These interactions play an important role in the membrane function[11,12] but have not been quantified in living cells, nor in realistic bilayer systems, such as liposomes or freestanding lipid bilayers. As an example, ion-specific effects that involve the complex interaction between cations, surface charges, polar, and hydrophobic groups are important in signaling through ion-specific channels in synapses[13–16], the folding of proteins[17,18], the formation of secondary structures[19,20], and the induction of cell death by the presence of charged phosphatidylserine lipids[21]. How these processes work on a molecular level is presently not understood. Although molecular level experiments on model planar air/lipid/water monolayer systems[22–25], and supported bilayer—protein systems[19,26], have revealed ingredients of the already complex molecular behavior that consists of different types of interactions between ion, lipids, proteins, and water molecules, they do not take into account the length scale, and the spatial and temporal dynamic behavior that is clearly important for the bilayer membranes of cells. Additionally, several molecular dynamics simulations have hinted on the importance of local and transient nature of model bilayer systems[22,27–32]. An important question, therefore, is whether the behavior of membranes and membrane hydration in particular can be captured, by averaging spatial and temporal fluctuations in the system[33]. Such averaging invokes assumptions that are necessarily made by the technical limitations that result in the application of mean-field model descriptions.

Here, we probe the interactions of divalent cations ($Ca^{2+}$, $Ba^{2+}$, and $Mg^{2+}$ at physiologically relevant concentrations), with water and negatively charged freestanding lipid bilayers using high-throughput wide-field second harmonic (SH) microscopy. We find that divalent cation–membrane interactions result in the formation of short-lived (<500 ms), ~1.5 micron-sized domains of orientationally ordered water. The SH intensity is converted into membrane potential ($\Phi_0$), surface charge density ($\sigma_0$), membrane hydration free energy ($\Delta G$), and ion binding dissociation constant ($K_D$) maps. The ion-induced changes follow the order $Ca^{2+} > Ba^{2+} > Mg^{2+}$ for all four quantities. The dissociation constant ($K_D$) of the domains reach values up to $2.7 \times 10^{-12}$ M, deviating up to four orders of magnitude from dissociation constant based on a mean-field interpretation. Using an electromechanical theory of membrane bending, we show that transient electric field gradients across the membrane lead to transient curvature fluctuations, resulting in the temporal and spatial fluctuations in membrane mechanical properties.

## Results and discussion

**Spatiotemporal ion-specific effects.** Freestanding horizontal planar lipid membranes were formed following the Montal-Müller method[34,35]. Two separate lipid monolayers at the air/water interface were apposed in an 80–120-μm-sized circular aperture in a 25-μm thick Teflon film. The horizontally mounted membrane was imaged with a medium repetition rate, wide-field nonlinear SH microscope. Two counter-propagating 190 fs, 1032 nm, 200 kHz pulsed beams with an opening angle of 45 deg illuminate the membrane interface, such that phase-matched SH photons are emitted and recorded along the membrane surface

normal (Fig. 1a, see refs. [36,37]). Figure 1b shows a SH image recorded of a symmetric lipid bilayer of identical leaflets composed of 70:30 mol% 1,2-diphytanoyl-sn-glycero-3-phosphocholine (DPhPC) and 1,2-diphytanoyl-sn-glycero-3-phosphate (DPhPA). Figure 1c shows a SH image recorded of an asymmetric lipid bilayer composed of a DPhPC leaflet (bottom leaflet) and a 70:30 mole% mixture of DPhPC:DPhPA (top leaflet). No coherent SH photons are generated by the symmetric bilayer while the asymmetric bilayer does generate a SH response. As we showed in ref. [37], the SH response arises from the charge–dipole interaction between the charged head groups and the dipolar water molecules, which creates a non-random orientational distribution of water dipoles along the surface normal. Figure 1d–f shows SH images of the same symmetric DPhPC:DPhPA bilayer system as in Fig. 1b, but instead of using an aqueous KCl solution in contact with both leaflets, we replaced the solution adjacent to the top leaflet with $(CaCl_2)_{aq}$, $(BaCl_2)_{aq}$, and $(MgCl_2)_{aq}$ solutions of the same ionic strength (150 μM).

Adding $Ca^{2+}$, $Ba^{2+}$, or $Mg^{2+}$ to the aqueous phase results in a spatially fragmented SH response. $Ca^{2+}$, $Ba^{2+}$, and $Mg^{2+}$ are known to interact specifically with negatively charged lipid head groups[22,23,31,38,39] forming lipid–cation complexes. The water in contact with such neutral cation–lipid clusters has a negligible orientational ordering along the interfacial normal[40]. Membrane water in contact with free head group charges, on the other hand, does exhibit an orientational order along the interfacial normal. When both structures are present on opposite sides of the membrane, centrosymmetry is broken, resulting in domains of bright SH intensity. The degree to which the centrosymmetry is broken depends on the strength of interaction between the divalent ion and the negatively charged head groups. Figure 1 shows that the number of domains and the relative intensity decreases in the order $Ca^{2+} > Ba^{2+} > Mg^{2+}$. This implies that the interaction of $Ca^{2+}$ with the negatively charged head groups is stronger when compared to $Ba^{2+}$ and $Mg^{2+}$.

To analyze the properties of these domains in more detail, we turn to single frame analysis (560 ms/frame) and analyze the normalized spatial (Fig. 2a, b) and temporal (Fig. 2c) correlations between the domains in a single leaflet, and the coupling between domains in both leaflets (Fig. 2d). The full-width at half maximum (FWHM) of the spatial correlation function reports on the characteristic radius of the domains, whereas the FWHM of the temporal autocorrelation function reports on the characteristic lifetime of the domains. The spatiotemporal evolution of the domains on a single leaflet is shown in Fig. 2a, where three consecutive time frames are shown (raw data), each recorded with a 560 ms integration time. Figure 2b shows the normalized spatial autocorrelation function (SACF) for the consecutive time frames of Fig. 2a. The gray curves are SACFs of the individual frames, the black data points are the average SACFs of the 20 frames fitted with a Gaussian curve. The average radius of the domains, derived from the FWHM of Fig. 2b is 1.5 μm. Figure 2c shows the normalized temporal autocorrelation function (TACF) for the consecutive time frames of Fig. 2a. Applying the same analysis to a series of single frame images of $Ba^{2+}$ and $Mg^{2+}$, we obtained the same average radius and temporal decay of their spatiotemporal correlations (Supplementary Note 1, Supplementary Fig. 1). Because the temporal correlation function decays faster than the recording time, the characteristic time of each domain is, therefore, shorter than the recording time. Thus, there is no correlation between the domains on the time scale of acquisition.

To understand the coupling between the SH domains on each leaflet, we recorded SH images of a 70:30 mole% DPhPC:DPhPA symmetric bilayer with CaCl₂ added to both leaflets at 150 μM ionic strength. The 20 frame (10.4 s) average result is shown in Fig. 2d.

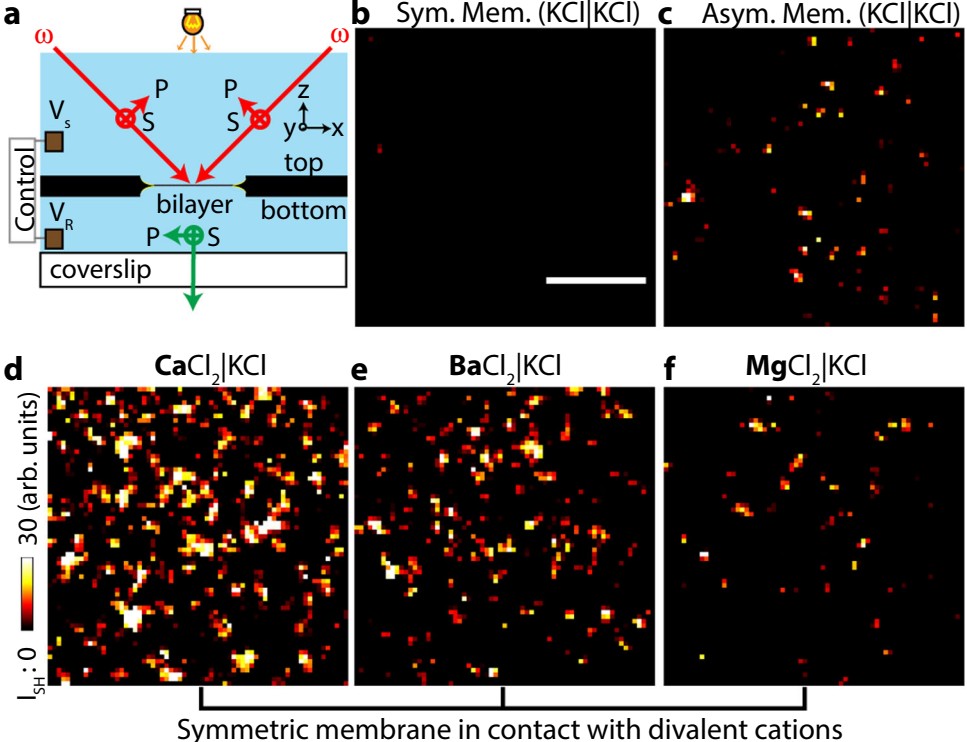

**Fig. 1 Divalent cations induce transient domains of ordered interfacial water. a** Two counter-propagating beams (190 fs, 1030 nm, ω, red arrows) overlap in space and time to illuminate the lipid bilayer membrane. SH photons (2ω, green arrow) are collected (magnification: 50×, NA = 0.65) in the phase-matched direction. SH images of **b** a symmetric membrane composed of 70:30 mol% DPhPC:DPhPA and **c** an asymmetric membrane composed of 70:30 mol% DPhPC:DPhPA (top leaflet), and DPhPC (bottom leaflet) in contact with a 150 μM pH neutral KCl solution. **d–f** SH images of a symmetric membrane composed of 70:30 mol% DPhPC:DPhPA with the bottom leaflet in contact with a 150 μM pH neutral KCl solution, and the top leaflet in contact with a pH neutral $CaCl_2$ **d**, $BaCl_2$ **e**, and $MgCl_2$ **f** at 150 μM ionic strength. The images were collected with all beams P-polarized, and represent 20 × 560 ms frame averages. The scale bar (10 μm) is the same for all images. The SH image of a 150-μM KCl solution was subtracted from the images to remove the hyper-Rayleigh scattering contribution.

The image shows a SH contrast that has a comparable intensity to the response of the asymmetric lipid bilayer in Fig. 1c, but less intensity compared to the response of a symmetric bilayer in Fig. 1d where only one leaflet is in contact with $CaCl_2$. Addition of $Ca^{2+}$ to both leaflets should lead to a vanishing SH response if neutral ion-lipid complexes are formed. This is especially true if the domains on opposing leaflets are in registry, i.e., the leaflets are strongly coupled[41]. However, if the leaflets are not in registry, i.e., not strongly coupled, then a non-vanishing SH response is expected. The non-vanishing SH response in Fig. 2d suggests that the domains in the opposing leaflets are not strongly coupled.

**Quantifying membrane potential and free energy changes**. Figures 1 and 2 show that divalent cations induce transient structural heterogeneities of ordered water in freestanding lipid bilayer membranes. The hydration shells of both leaflets are only partially correlated, and specific ion effects are important. To obtain more insight into the physicochemical behavior, we next quantify the spatiotemporal membrane potential, surface charge distribution, and free energy landscape. Theory and experiments[42,43] have shown that the SH intensity of an interface depends quadratically on the surface potential ($\Phi_0$) and for lipid bilayers with two oppositely oriented membrane interfaces we have[37]:

$$I(2\omega, x, y) \sim I(\omega, x, y)^2 \left| \chi_{s1}^{(2)}(x,y) - \chi_{s2}^{(2)}(x,y) + f_3 \chi^{(3)'}(\Phi_{0,1}(x,y) - \Phi_{0,2}(x,y)) \right|^2 \quad (1)$$

where $\chi_{s,1}^{(2)} / \chi_{s,2}^{(2)}$ and $\Phi_{0,1} / \Phi_{0,2}$ are the surface second-order susceptibilities and the surface potentials of each leaflet,

respectively, and $\chi^{(3)'}$ is an effective third-order susceptibility of the aqueous phase[44]. The subscripts 1 and 2 refer to the top/bottom leaflets of the bilayer, $x$ and $y$ are the spatial coordinates, $\omega$ is the frequency of the fundamental beam, and $f_3$ is an interference term, where $f_3 \rightarrow 1$ for transmission experiments[37,44,45]. For symmetric bilayers, $\chi_{s1}^{(2)} = \chi_{s2}^{(2)}$, and Eq. (1) is reduced to $I(2\omega, x, y) \propto \left| \chi^{(3)'}(\Phi_{0,1}(x,y) - \Phi_{0,2}(x,y)) \right|^2$. With identical surface potentials the coherent SH intensity vanishes (Fig. 1b). By recording SH images of an asymmetric charged leaflet as a function of external electric bias, we showed that it is possible to convert the SH intensity scale into a membrane potential scale, $\Delta\Phi_0 = \Phi_{0,1}(x,y) - \Phi_{0,2}(x,y)$ (Supplementary Note 4). From $\Delta\Phi_0$, the change in electrostatic free energy ($\Delta G$) is found as $\Delta G = 2e\Delta\Phi_0$, and the ion–lipid dissociation constant ($K_D$) is given by $\Delta G = -RTln(K_D)$ where $T$ is the temperature and $R$ is the gas constant[46]. Furthermore, the surface charge density ($\Delta\sigma_0$) was modeled with a parallel plate capacitor in contact with aqueous solution[47], appropriate for divalent ion negatively charged interface interactions[22,31], where $\Delta\sigma_0 = C \times \Delta\Phi_0$, with $C = \varepsilon_0 \epsilon / d$, $\epsilon = 2.1$, and $d = 4$ nm, the dielectric constant, and thickness of the hydrophobic core respectively[35].

Figure 3 shows the measured intensity values (Fig. 3a; corrected for hyper-Rayleigh scattering (HRS) by image subtraction), extracted values for the membrane potential ($\Delta\Phi_0$) and the surface charge density ($\Delta\sigma_0$; Fig. 3b), the electrostatic free energy change ($\Delta G$) and the binding dissociation constant ($K_D$; Fig. 3c) for the different divalent cations SH imaged in Fig. 1d–f. The

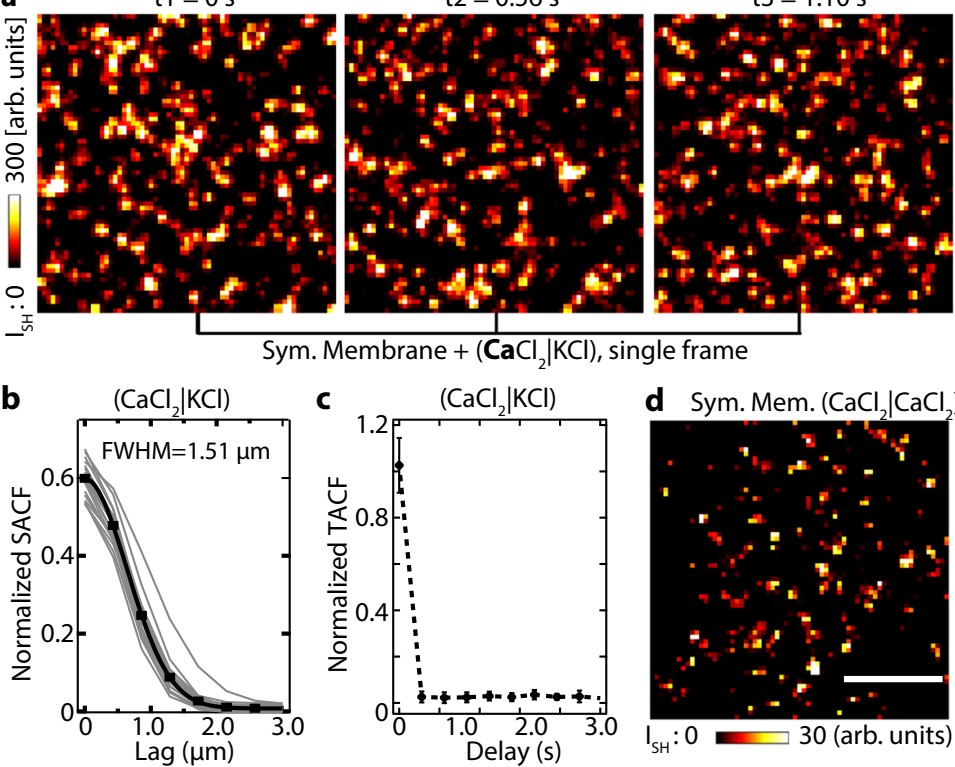

**Fig. 2 Spatiotemporal dynamics of ion-induced ordered water domains. a** Time series of SH images (560 ms each) of a symmetric membrane composed of 70:30 mol% DPhPC:DPhPA with the top leaflet in contact with $(CaCl_2)_{aq}$ and the bottom leaflet in contact with $(KCl)_{aq}$. **b** Normalized spatial autocorrelation functions (SACF) of the single frame images in **a**. The gray curves represent the SACF of the individual frames (20 frames total) and the black data points are the average SACF values of all the gray curves fitted with a Gaussian curve. **c** Normalized TACF of the images in **a** showing no temporal correlation between frames. **d** SH image (average of 20 frames, 560 ms) of a symmetric membrane composed of 70:30 mol% DPhPC:DPhPA where both leaflets are in contact with $(CaCl_2)_{aq}$ with the same 150 μM ionic strength. This SH image was corrected for hyper-Rayleigh scattering by subtracting a 150-μM KCl solution SH image. All images were collected with all beams P-polarized. The scale bar (10 μm) is the same for all images.

average quantities are displayed on each panel. Table 1 lists the image stack averaged values of ($\Delta\Phi_0$, $\Delta\sigma_0$, $\Delta G$, and $K_D$ and the values for the domains (corrected for HRS; Supplementary Note 2, Supplementary Note 3, Supplementary Figs. 2 and 3). Our average values are in good agreement with (the limited) literature on binding constants[48–51]. However, Figs. 1–3 and Table 1 show a very interesting unexpected aspect: instead of a uniformly distributed divalent cation–lipid binding, there are transient structures. Examining these domains, much larger values are found for $\Delta\Phi_0$, $\Delta\sigma_0$, $\Delta G$, and $K_D$, resulting in actual binding dissociation constants that are up to four orders of magnitude larger than the total spatially averaged value. In addition, there are places on the image where virtually no binding occurs, and the chemical structures where binding occurs are short lived and continuously redistribute across the membrane.

**From transient membrane structure to curvature**. The transient structural domains (Fig. 3) exhibit membrane potential fluctuations of up to −386 mV (with $\Delta G = 28.6$ kT). Although the molecular level interactions are complex, our findings can be rationalized qualitatively as follows (Fig. 4): the addition of divalent ions leads to an electrostatic field gradient (Fig. 4b) that induces strain in the membrane via a surface pressure gradient across the membrane and steric pressure along the membrane[52,53]. A homogeneous distribution of ions results in high local strain due to the electromechanical coupling with membrane fluctuations. However, the local clustering of ions can potentially relax high strains when these clusters are spread out

over larger distances[54,55] (Fig. 4c). Using a mean-field liquid crystal membrane approximation to calculate the curvature as a function of an applied electric field[56,57], and using again the approximation of an electric capacitor, we estimated the curvature $H$ for the three cations. The total curvature of the membrane in response to an applied electric field ($E$) is given by $H = \frac{fE}{2\kappa} = \frac{f\Delta\Phi_0}{2\kappa d}$, with $f$ the flexocoefficient of the membrane, and $\kappa$ the bending modulus. Using $10$ kT $< \kappa < 20$ kT and $10^{-21} < f < 10^{-18}$ C[11], we find $3.6 \times 10^{-4} < H < 0.98$ nm$^{-1}$. The small value of curvature corresponds to a membrane with a large bending modulus and with a low flexoelastic coefficient, while the large curvature values correspond to a membrane with a small bending modulus and a large flexoelastic coefficient. This model ignores thermal fluctuations, cation penetration into the headgroup[22,27–30], and other electromechanical effects that should be taken account for quantitative analysis of curvature fluctuations. Nevertheless, it shows that the measured transient fluctuations in Fig. 3 can lead to transient curvature fluctuations. This, in turn, will result in surface tension fluctuations[58,59]. Figure 4d shows topographic maps of membrane deformation generated from the center of the images of Fig. 3b, showing that different electric fields can induce different extents of transient curvatures for Ca$^{2+}$, Ba$^{2+}$, and Mg$^2+$ ions, following the trend of the Hofmeister series. The large and dense potential fluctuations induced by Ca$^{2+}$ should thus result in a larger variation in the height profile of the membrane. The smaller and more spread out fluctuations for Mg$^{2+}$ (and Ba$^{2+}$) result in smaller curvature deviations and thus in smaller height profile fluctuations.

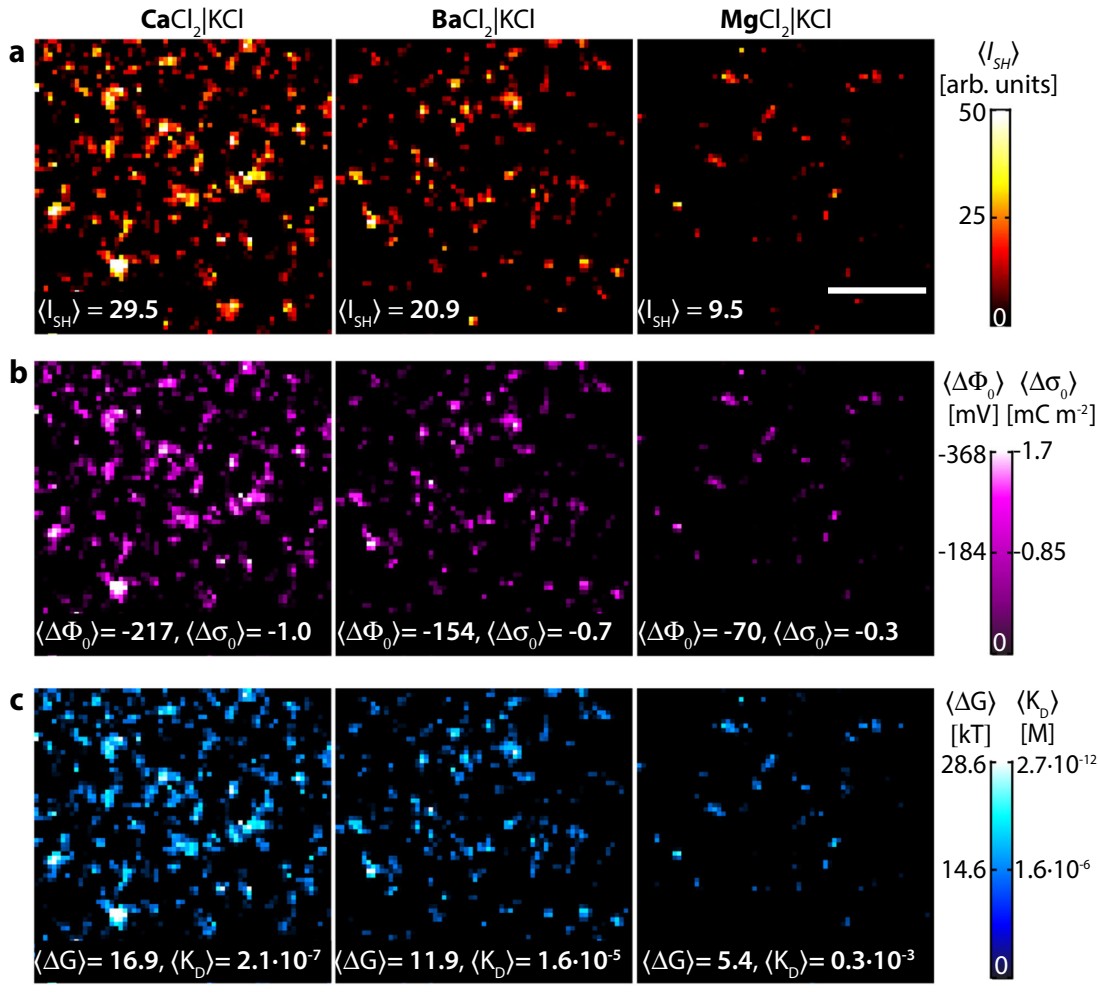

**Fig. 3 Quantifying the free energy landscape of membranes. a** Average SH intensity $\langle I \rangle$, **b** the change in membrane potential $\langle \Delta \Phi_O \rangle$, and surface charge density $\langle \Delta \sigma_O \rangle$ (assuming a parallel plate capacitor model), **c** free energy of binding $\langle \Delta G \rangle$ and ion–lipid dissociation constant $\langle \Delta K_D \rangle$ for symmetric membranes composed of 70:30 mol% DPhPC:DPhPA where the top leaflet is in contact with divalent cations: $Ca^{2+}$, $Ba^{2+}$, and $Mg^{2+}$, and the bottom leaflet is in contact with KCl ions with the same ionic strength (150 µM). Units are provided with the color scale to the right. The scale bar (10 µm) is the same for all images.

**Table 1 Image-averaged and single-domain values of membrane potential ($\Delta \Phi_O$), surface charge density ($\Delta \sigma_O$), electrostatic free energy of binding ($\Delta G$), and ion–lipid dissociation constant ($K_D$).**

|  |  | $\Delta \Phi_O$ (mV) | $\Delta \sigma_O$ (mC m$^{-2}$) | $\Delta G$ (kT) | $K_D$ (M)* |
|---|---|---|---|---|---|
| $Ca^{2+}$ | Average | −217 | −1.02 | 17 | $2.17 \times 10^{-7}$ |
|  | Domain | $245 < |\Delta \Phi_O| < 329$ | $1.15 < |\Delta \sigma_O| < 1.55$ | $19 < \Delta G < 26$ | $1.9 \times 10^{-8} < K_D < 1.3 \times 10^{-11}$ |
| $Ba^{2+}$ | Average | −154 | −0.72 | 12 | $1.6 \times 10^{-5}$ |
|  | Domain | $130 < \Delta \Phi_O| < 209$ | $0.61 < |\Delta \sigma_O| < 0.98$ | $10 < \Delta G < 16$ | $2.4 \times 10^{-4} < K_D < 1.15 \times 10^{-7}$ |
| $Mg^{2+}$ | Average | −70 | −0.33 | 5.4 | $2.3 \times 10^{-2}$ |
|  | Domain | $68 < |\Delta \Phi_O| < 108$ | $0.32 < |\Delta \sigma_O| < 0.51$ | $5.3 < \Delta G < 8.4$ | $2.0 \times 10^{-2} < K_D < 4.51 \times 10^{-3}$ |

The range of the domain values is taken as domain average ± 1 standard deviation.
*The following are found in the literature: $Ba^{2+}$ with 1,2-dimyristoyl-sn-glycero-3-phosphate (DMPA), $K_D = 10^{-6}$ M (ref. [48]), and $Ca^{2+}$ induces fusion of phosphatidic acid (PA) containing vesicles at -100 µM (ref. [49–51]).

In summary, we demonstrate that at physiological concentrations $Ca^{2+}$, $Ba^{2+}$, and $Mg^{2+}$ induce short-lived (<500 ms) ~1.5 micron-sized domains of ordered interfacial water. Converting the SH intensity into membrane potential, surface charge density, membrane hydration free energy, and binding dissociation constant maps, we obtain trends in the order $Ca^{2+} > Ba^{2+} > Mg^{2+}$, for all four quantities and reach domain values of −368 mV, −1.7 mC/m², 28.6 kT, and $2.7 \times 10^{-12}$ M that deviate up to four orders of magnitude from current reaction constant values that are based on a mean-field interpretation. Additionally, the

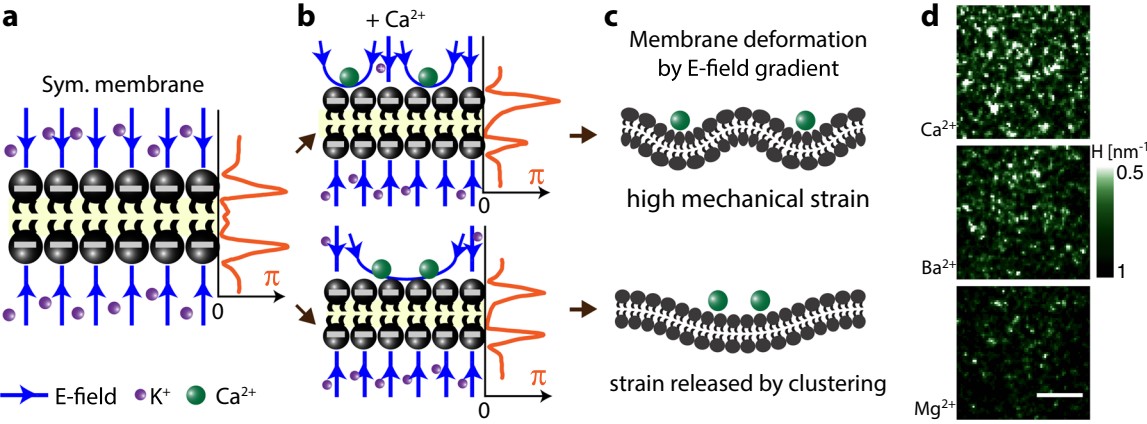

**Fig. 4 Specific ion-induced transient curvature. a** Schematic of a symmetric negatively charged membrane in solution. **b** The binding of Ca$^{2+}$ to a symmetric negatively charged membrane induces an asymmetric electric field gradient, which is different for a homogeneous (top) or clustered (bottom) distribution of cations, leading to different pressure profiles, and **c** consequently different membrane curvatures. **d** Topographic maps of membrane deformation of the membranes of Fig. 3, showing more pronounced transient curvature fluctuations for Ca$^{2+}$ > Ba$^{2+}$ > Mg$^{2+}$ (using $f = 5 \times 10^{-19}$ C, $\kappa = 15$ kT, and $d = 4$ nm). The scale bar is 10 μm.

transient electric field gradients across the membrane lead to transient curvature, resulting in temporal and spatial variations in the mechanical properties for the membrane. Although we used a mean-field model as a first approximation for curvature estimation, there have been many molecular dynamics simulations of cation interaction with phospholipid bilayers[22]. These simulations have shown that the cations can form large scale clusters that can dehydrate and neutralize anionic lipid bilayers[22], strongly adsorb onto the lipid bilayer[22,27–30], and compress the lipid bilayer laterally[22]. Additionally, calcium ions are known to penetrate deeply into the lipid bilayer in a concentration-dependent manner[22]. Here, we show that the interactions of cations with the membrane also induce curvature of the membrane. Thus, aside from having a local-specific interaction with lipids as has been previously known[22–25], divalent ions also influence the spatiotemporal chemical, electric and mechanical membrane properties, leading to a diversification of membrane environments, and a new mechanism for coupling local chemical interactions with macroscopic behavior. Such an effect potentially plays an important role in membrane protein interactions, important for structuring, signaling, and transport.

## Methods
**Chemicals**. DPhPC and DPhPA in powder form (>99%, Avanti Polar Lipids, Alabama, USA), hexadecane (C$_{16}$H$_{34}$, 99.8%, Sigma-Aldrich), hexane (C$_6$H$_{14}$, >99%, Sigma-Aldrich), chloroform (>99.8%, Merck), hydrogen peroxide (30%, Reactolab SA), sulfuric acid (95–97%, ISO, Merck), KCl (99.999%, Aros), CaCl$_2$ (99.999%), MgCl$_2$ (99.99%), and BaCl$_2$ (99.999%, Sigma-Aldrich) were used as received. All aqueous solutions were made with ultra-pure water (H$_2$O, Milli-Q UF plus, Millipore, Inc., electrical resistance of 18.2 MΩ cm). All aqueous solutions were filtered with 0.1 μM Millex filters. The coverslips used in the imaging were pre-cleaned with piranha solution (1:3–30% H$_2$O$_2$: 95–97% H$_2$SO$_4$) and thoroughly rinsed with ultra-pure water.

**Formation of freestanding horizontal planar lipid bilayers**. Freestanding horizontal planar lipid bilayers were formed following the procedure of Montal-Müller[34,60]. Two separated lipid monolayers on an air/water interface were combined in a ~80–120-μm aperture in 25-μm thick Teflon film. The presence of a bilayer was confirmed with white light imaging and electrical recordings with specific capacitance, $C_m > 0.7$ μF/cm$^2$, specific resistance, $R_m \sim 10^8$ Ω cm$^2$ (refs. [61,62]). The composition of the leaflets and the aqueous solution where the bilayer leaflets reside are controllable in situ. Unless stated, all measurements were performed at pH neutral conditions.

**Electrical characterization of freestanding lipid membranes**. Ag/AgCl pellet electrodes were placed on each side of the bilayer and electrical measurements were recorded through the HEKA patch clamp amplifiers. Capacitance and resistance

measurements were made with HEKA's built-in software-based lock-in amplifier[63]. For more details, see ref. [37].

**SH imaging**. The imaging setup has been characterized in detail in refs. [36,37,64] based on principles of SH scattering[65]. Two counter-propagating beams from a Yb: KGW femtosecond laser (Light Conversion Ltd) delivering 190 fs pulses, 1028 nm with a 200 kHz repetition rate were incident at 45° with respect to the membrane. Each beam was loosely focused using an $f = 20$ cm doublet lens (B coating, Thorlabs), and polarization controlled using a linear polarizer (Glan-Taylor polarizer, GT10-B, Thorlabs) and a zero-order λ/2 wave plates (WPH05M-1030, Thorlabs). The average power for each arm was set to ~110 mW. The phase-matched SH photons were collected with a 50× objective lens (Mitutoyo Plan Apo NIR HR Infinity-Corrected Objective, 0.65 NA in combination with a tube lens (Mitutoyo MT-L), a 900 nm short pass filter (FES0900, Thorlabs), a 515 nm band-pass filter (FL514.5-10), and an intensified electronically amplified CCD camera (IE-CCD, PiMax4, Princeton Instruments). A 400 mm meniscus lens was placed behind the objective lens to remove spherical aberrations induced by the coverslip. The transverse resolution, and thus the pixel width was 430 nm. All images were recorded with the beams polarized parallel to the plane of incidence (P). The acquisition time of the images was 560 ms.

## Data availability
The data that support the findings of this study are available from the corresponding author upon reasonable request.

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

## Acknowledgements

We would like to thank N. Levinger, G. Pabst, P. Wittung-Stafshed, P. Pohl, and E. Yan for useful discussions. This work is supported by the Julia Jacobi Foundation, the Swiss National Science Foundation (grant number 200021-140472), and the European Research Council grant 616305.

## Author contributions

O.B.T. performed the experiments, O.B.T. performed the analysis, O.B.T., H.I.O., P.R, and S.R wrote the manuscript. S.R. supervised and conceived the project. O.B.T., H.I.O., and S.R designed the experiments, and P.R. conducted the modeling analysis.

## Competing interests

The authors declare no competing interests.
