## [Peer Review File · Communications Chemistry]

Reviewers' comments:

Reviewer #1 (Remarks to the Author):

Tarun and co-authors study the transient binding of Ca^{2+} , Ba^{2+} and Mg^{2+} cations to neutral and negatively-charged phospholipid membranes. They use free-standing planar bilayers of both symmetric and asymmetric lipid composition and apply second harmonic imaging. They detect local binding of cations and formation of transient (<500 ms) microdomains (~ 1.5 μm). They interpret some of the observation using local curvature and membrane bending induced by the presence of adsorbed cations.

It is an interesting work tackling the issue of locality and transient character of ion-membrane interactions, which is crucial for understanding, e.g., Ca^{2+} behavior close to the inner cell membrane leaflet during calcium signaling. The work is well executed and clearly presented. Nevertheless, I found several major and minor issues that should be addressed before publication:

Major:

1. Curvature considerations do not take into account the penetration of cations into the headgroup region. As shown via MD (e.g., Ref. 22), membrane-water interface, from the point of view of cation binding, is far from being a plane on the nanometer length scale. This limitation of the presented bending considerations should be pointed out.

2. Along similar lines, both Introduction and Discussion seem to omit MD simulation studies of model bilayer systems. Despite all limitations, MD is the technique addressing the local and transient character of cation-membrane interactions directly, and several relevant papers have been published on this subject recently.

3. The title claims "domains of ordered water" whereas this is not a central conclusion of the presented results. Furthermore, based on the given experimental data, the membrane bending is only a qualitative hypothesis. Instead, the data show transient local binding of cations to negatively charged membranes. Hence, I strongly suggest modifying the title and, accordingly, some parts of the manuscript.

Minor:

A. Why PA lipids and not PS. The latter is more relevant to mammalian cells. Please, comment on this.

B. How/whether the transient character of the formed domains is accounted for in the mean-field approximation used for curvature considerations?

C.p. 3: "80-120- μm -sized circular aperture" vs. p. 10: "80-100- μm aperture"?

Reviewer #2 (Remarks to the Author):

This manuscript reports on the transient binding of divalent cations onto free standing lipid bilayers as observed through the induced ordering of the water network by second harmonic imaging.

It is observed that short-lived (<500 ms), 1.5 μm sized domains are formed with an increasing density from Mg^{2+} to Ba^{2+} to Ca^{2+} . The domains are caused by divalent cations binding to the lipid membrane causing an imbalance in the charge distribution across the membrane, which orient the water dipoles giving rise to the second harmonic response. The observation is validated by comparing the second harmonic response from asymmetric membranes and symmetric

membranes in contact with different electrolyte solutions on either side of the membrane (one monovalent and one divalent). However, the domain formation is also found in symmetrically solvated symmetric membranes due to the transient nature of the binding causing a transient imbalance in the spatially distributed surface potential.

The observation of individual binding domains and their special and temporal characterization is very interesting and will be interesting to a broad community appropriate for Nature Communications Chemistry. Particularly in the light of the label free method, which is a very clever application of second harmonic imaging to probe ion binding through the associated water alignment.

However, while the content of the manuscript is very appropriate for Nature Communications Chemistry, the wording throughout the manuscript could be improved to further the clarity of the arguments to reach the broad audience. This starts with the abstract, which is not sufficiently clear.

Specifically, there are a number of statements, which are not clear:

On Page 2:

"The ion-induced changes follow the order $\text{Ca}^{2+} > \text{Ba}^{2+} > \text{Mg}^{2+}$, for all four quantities and reach domain values of -368 mV, -1.7 mC/m², 28.6 kT, and $2.7 \cdot 10^{-12}$ M, with KD deviating up to 4 orders of magnitude from current values based on a mean-field interpretation."

The wording of this needs to be improved.

Also on Page 2:

"Additionally, the transient electric field gradients across the membrane lead to transient curvature, resulting in temporal and spatial fluctuations in the mechanical properties of the membrane."

This is an interpretation of the data and should be described as such.

On Page 3:

"Fig. 1 shows that the number of domains and the relative intensity decreases in the order $\text{Ca}^{2+} > \text{Ba}^{2+} > \text{Mg}^{2+}$."

This sentence needs to be followed with a statement of what this implies.

On Page 4:

"Applying the same analysis to a series of single frame images of Ba^{2+} and Mg^{2+} , we obtained the same average radius and temporal decay of their spatiotemporal correlations (see SI, S1, Fig. S1). Thus, there is no correlation between the domains on the time scale of acquisition."

I do not understand how the authors arrived at this conclusion.

On Page 5

First the authors say:

"This means that there is no strong coupling between the hydrations shells of opposing leaflets"

And then:

“The hydration shells of both leaflets are only partially correlated”

It is not sufficiently explained how the authors came to these conclusions. Presumably this is from a comparison of the intensities of the symmetric and asymmetric bilayers but the intensities are not quantified, just stating that:

“the response of the asymmetric lipid bilayer in Fig. 1c but less intensity compared to the response of a symmetric bilayer”

How do the intensities compare and what would be the limit of what would be considered strong or partial coupling?

Lastly on page 5: “These are clearly surprising findings...”. I would be careful with such a statement.

The manuscript ends with briefly discussing a membrane bending model to qualitatively rationalize the findings. This is outside my expertise and is not sufficiently described for me to evaluate. The model is also not well connected with the rest of the paper and not mentioned earlier. However, this model could potentially explain some of the initial questions I had about the findings. That the domains are associated with clusters of ions and not single ion bindings, which could explain the size of the domains and potentially also the magnitude of the second harmonic signal. Could the magnitude of the second harmonic signal be used to estimate the number of ions in a domain? If the authors are confident in this interpretation, then this model should be mentioned earlier, say in the abstract, to provide a framework for understanding the results while reading the paper.

In summary, the manuscript presents some very interesting and unique results that are appropriate for Nature Communications Chemistry, but the language and arguments need to be tightened up to better explain the findings and conclusions.

We would like to thank the referees for carefully reading our manuscript and for providing us with valuable comments and suggestions to improve the manuscript. We have revised the manuscript on the basis of their comments and we present our detailed replies below. The comments of the referee are in *italic* and our responses in normal font.

Reviewers Comments

Reviewer #1 (Remarks to the Author):

Tarun and co-authors study the transient binding of Ca²⁺, Ba²⁺ and Mg²⁺ cations to neutral and negatively-charged phospholipid membranes. The use free-standing planar bilayers of both symmetric and asymmetric lipid composition and apply second harmonic imaging. The detect local binding of cations and formation of transient (<500 ms) microdomains (~1.5 um). They interpret some of the observation using local curvature and membrane binding induced by the presence of adsorbed cations.

It is an interesting work tackling the issue of locality and transient character of ion-membrane interactions, which is crucial for understanding, e.g., Ca²⁺ behavior close to the inner cell membrane leaflet during calcium signaling. The work is well executed and clearly presented. Nevertheless, I found several major and minor issues that should be addressed before publication:

Major:

1. Curvature considerations do not take into account the penetration of cations into the headgroup region. As shown via MD (e.g., Ref. 22), membrane-water interface, from the point of view of cation binding, is far from being a plane on the nanometer length scale. This limitation of the presented bending considerations should be pointed out.

The referred paper uses PS and PC lipids while we mostly focus on PA lipids on this paper. Nevertheless, we use the mean-field liquid crystal approximation to provide a qualitative framework to show that transient electric field gradient across the bilayer leads to transient curvature fluctuations. It ignores thermal fluctuations and other electromechanical effects but nonetheless sufficient for qualitative analysis. A quantitative analysis of curvature fluctuations should take into account thermal fluctuations as well as cation penetration into the headgroup region. In the discussion section, we included the limitations of the model. "...This model ignores thermal fluctuations, cation penetration into the headgroup and other electromechanical effects that should be taken account for quantitative analysis of curvature fluctuations. Nevertheless, it shows that the measured transient fluctuations in Fig. 3 can lead to transient curvature fluctuations."

We also discussed the limitation in the conclusions section. "...Although we used a mean-field model as a first approximation for curvature estimation, there have been many molecular dynamics simulations of cation interaction with phospholipid bilayers²². These simulations have shown that the cations can form large scale clusters that can dehydrate and neutralize anionic lipid bilayers²², strongly adsorb onto the lipid bilayer^{22,27-30}, and compress the lipid bilayer laterally²². Additionally, calcium ions are known to penetrate deeply into the lipid bilayer in a concentration dependent manner²². Here, we show that the interactions of cations with the membrane also induce curvature of the membrane."

2. Along similar lines, both Introduction and Discussion seem to omit MD simulation studies of model bilayer systems. Despite all limitations, MD is the technique addressing the local and transient character of cation-membrane interactions directly, and several relevant papers have been published on this subject recently.

Thank you for the comment. As noted above, we cited several related MD studies in the introduction and the discussion section.

3. The title claims "domains of ordered water" whereas this is not a central conclusion of the presented results.

Furthermore, based on the given experimental data, the membrane bending is only a qualitative hypothesis. Instead, the data show transient local binding of cations to negatively charged membranes. Hence, I strongly suggest modifying the title and, accordingly, some parts of the manuscript.

We changed the title to “Transient domains of ordered water induced by divalent ions lead to lipid membrane curvature fluctuations”. We also mentioned the model in the abstract to provide a framework for understanding the experimental results.

Minor:

1. Why PA lipids and not PS. The latter is more relevant to mammalian cells. Please, comment on this.

We observe experimentally that PA lipids generate stronger SH intensity compared to PS in the presence of Ca^{2+} , Ba^{2+} and Mg^{2+} . This suggests that PA lipids have higher binding affinity compared to PS with respect to these divalent ions. This result is consistent with MD¹ and experiments on fusion of vesicles induced by divalent cations as well as calorimetric measurements of cations binding to negative membranes²⁻⁵. The binding sites for the PA head-group are exposed compared to PS with the serine molecule blocking the phosphate region. Furthermore, the conic structure of PA is more favourable for binding. These effects lead to better affinity for PA compared to PS. Nonetheless, it is still possible to use PS lipids and we have done so. It will be the subject of future studies.

2. How/whether the transient character of the formed domains is accounted for in the mean-field approximation used for curvature considerations?

The curvature calculations assumed equilibrium of the stresses acting on the membrane (this is standard practice in the field) and the transient nature of the domains is introduced into model through the transient electric field gradient. At any instant in time, the given electric field gradient was assumed to impose stresses on the membrane that equilibrated rapidly. In this work, our goal was qualitatively explain the transient curvature effects; there was no attempt at introducing the transient nature of the domains starting on the molecular level. Ongoing effort in a collaboration between the Rangamani and the Roke groups is focused on including the molecular effects and the transients.

C.p. 3: “80-120- μm -sized circular aperture” vs. p. 10: “80-100- μm aperture”?

Thank you for the comment. It should be 80-120- μm -sized circular aperture, we have changed the text accordingly

Reviewer #2 (Remarks to the Author):

This manuscript reports on the transient binding of divalent cations onto free standing lipid bilayers as observed through the induced ordering of the water network by second harmonic imaging.

It is observed that short-lived (<500 ms), 1.5 μm sized domains are formed with an increasing density from Mg^{2+} to Ba^{2+} to Ca^{2+} . The domains are caused by divalent cations binding to the lipid membrane causing an imbalance in the charge distribution across the membrane, which orient the water dipoles giving rise to the second harmonic response. The observation is validated by comparing the second harmonic response from asymmetric membranes and symmetric membranes in contact with different electrolyte solutions on either side of the membrane (one monovalent and one divalent). However, the domain formation is also found in symmetrically solvated symmetric membranes due to the transient nature of the binding causing a transient imbalance in the spatially distributed surface potential.

The observation of individual binding domains and their special and temporal characterization is very interesting and will be interesting to a broad community appropriate for Nature Communications Chemistry. Particularly in the light of the label free method, which is a very clever application of second harmonic imaging to probe ion binding through the associated water alignment.

However, while the content of the manuscript is very appropriate for Nature Communications Chemistry, the wording throughout the manuscript could be improved to further the clarity of the arguments to reach the broad audience. This starts with the abstract, which is not sufficiently clear.

Specifically, there are a number of statements, which are not clear:

On Page 2:

“The ion-induced changes follow the order $\text{Ca}^{2+} > \text{Ba}^{2+} > \text{Mg}^{2+}$, for all four quantities and reach domain values of -368 mV, -1.7 mC/m², 28.6 kT, and $2.7 \cdot 10^{-12}$ M, with K_D deviating up to 4 orders of magnitude from current values based on a mean-field interpretation.”

The wording of this needs to be improved.

Thank you for the comment. We have simplified the wording to: “The ion-induced changes follow the order $\text{Ca}^{2+} > \text{Ba}^{2+} > \text{Mg}^{2+}$ for all four quantities. The dissociation constant (K_D) of the domains reach values up to $2.7 \cdot 10^{-12}$ M, deviating up to 4 orders of magnitude from dissociation constant based on a mean-field interpretation.”

Also on Page 2: “Additionally, the transient electric field gradients across the membrane lead to transient curvature, resulting in temporal and spatial fluctuations in the mechanical properties of the membrane.”

This is an interpretation of the data and should be described as such.

Thank you for the comment, we changed the wording to “Using an electromechanical theory of membrane bending, we show that transient electric field gradients across the membrane lead to transient curvature fluctuations, resulting in temporal and spatial fluctuations in the mechanical properties of the membrane.”

On Page 3:

“Fig. 1 shows that the number of domains and the relative intensity decreases in the order $\text{Ca}^{2+} > \text{Ba}^{2+} > \text{Mg}^{2+}$.” This sentence needs to be followed with a statement of what this implies.

Thank you for the comment. We changed the paragraph:

Adding Ca^{2+} , Ba^{2+} or Mg^{2+} to the aqueous phase...resulting in domains of bright SH intensity. “The degree to which the centrosymmetry is broken depends on the on the strength of interaction between the divalent ion and the negatively charged head groups. Fig. 1 shows that the number of domains and the relative intensity decreases in the order $\text{Ca}^{2+} > \text{Ba}^{2+} > \text{Mg}^{2+}$. This implies that the interaction of Ca^{2+} with the negatively charged head groups is stronger when

compared to Ba²⁺ and Mg²⁺.

On Page 4:

“Applying the same analysis to a series of single frame images of Ba²⁺ and Mg²⁺, we obtained the same average radius and temporal decay of their spatiotemporal correlations (see SI, S1, Fig. S1). Thus, there is no correlation between the domains on the time scale of acquisition.”

I do not understand how the authors arrived at this conclusion.

Thank you for the comment. We clarified the paragraph by describing in more detail the spatial and temporal correlation functions. We inserted in the paragraph:

To analyze the properties of these domains ... and the coupling between domains in both leaflets (Fig. 2d). “The full width at half maximum (FWHM) of the spatial correlation function reports on the characteristic radius of the domains whereas the FWHM of the temporal autocorrelation function reports on the characteristic lifetime of the domains”... we obtained the same average radius and temporal decay of their spatiotemporal correlations. “Because the temporal correlation function decays faster than the recording time, the characteristic time of each domain is, therefore, shorter than the recording time. Thus, there is no correlation between the domains on the time scale of acquisition.”

On Page 5

First the authors say:

“This means that there is no strong coupling between the hydrations shells of opposing leaflets”

And then:

“The hydration shells of both leaflets are only partially correlated”

It is not sufficiently explained how the authors came to these conclusions. Presumably this is from a comparison of the intensities of the symmetric and asymmetric bilayers but the intensities are not quantified, just stating that:

“the response of the asymmetric lipid bilayer in Fig. 1c but less intensity compared to the response of a symmetric bilayer”

How do the intensities compare and what would be the limit of what would be considered strong or partial coupling?

Thank you for the comments. We have clarified what we meant by coupling. We agree that we could not completely quantify the coupling; at best we could say that it is not strongly coupled as oppose to partial coupling that suggest some degree of coupling. We inserted in the paragraph: To understand the coupling between ... where only one leaflet is in contact with CaCl₂. “Addition of Ca²⁺ on both leaflets, that forms neutral ion-lipid complex should lead to vanishing SH response. This is especially true if the domains on opposing leaflets are in registry, i.e., the leaflets are strongly coupled. However, if the leaflets are not in registry, i.e., not strongly coupled, then a non-vanishing SH response is expected. The non-vanishing SH response in Fig. 2d suggests that the domains in the opposing leaflets are not strongly coupled.”

Lastly on page 5: “These are clearly surprising findings...”. I would be careful with such a statement.

We removed this statement.

The manuscript ends with briefly discussing a membrane bending model to qualitatively rationalize the findings. This is outside my expertise and is not sufficiently described for me to evaluate. The model is also not well connected with the rest of the paper and not mentioned earlier. However, this model could potentially explain some of the initial questions I had about the findings. That the domains are associated with clusters of ions and not single ion bindings, which could explain the size of the domains and potentially also the magnitude of the second harmonic signal. Could the magnitude of the second harmonic signal be used to estimate the number of ions in a domain? If the authors are confident in this interpretations, then this model should be mentioned earlier, say in the abstract, to provide a framework for understanding the results while reading the paper.

The SH intensity reflects domains of hydration shells, i.e., ordered domains of water. The

ordering of water can be induced by cluster of ions. However, to state explicitly that we observe ion-membrane clusters is an indirect qualitative finding and does not directly follow from the result. We can only say that the charge density is higher leading to stronger water ordering and thus higher SH intensity. It could be that further future analysis may allow us to quantify the number of ion – lipid pairs.

In summary, the manuscript presents some very interesting and unique results that are appropriate for Nature Communications Chemistry, but the language and arguments need to be tightened up to better explain the findings and conclusions.

Thank you for the very helpful comments. We have improved the abstract by including the model to provide a framework for understanding the experimental results. Furthermore, we improved the language and presentation throughout the manuscript.

References

1. Faraudo, J. and A. Travestet (2007). "Phosphatidic Acid Domains in Membranes: Effect of Divalent Counterions." *Biophysical Journal* 92(8): 2806-2818.
2. Papahadjopoulos, D., W. J. Vail, W. A. Pangborn and G. Poste (1976). "Studies on membrane fusion. II. Induction of fusion in pure phospholipid membranes by calcium ions and other divalent metals." *Biochimica et Biophysica Acta (BBA) - Biomembranes* 448(2): 265-283.
3. Ito, T. and S.-I. Ohnishi (1974). " Ca^{2+} -induced lateral phase separations in phosphatidic acid-phosphatidylcholine membranes." *Biochimica et Biophysica Acta (BBA) - Biomembranes* 352(1): 29-37
4. Ohki, S. and K. Arnold (2000). "A mechanism for ion-induced lipid vesicle fusion." *Colloids and Surfaces B: Biointerfaces* 18(2): 83-97.
5. Garidel, P. and A. Blume (2000). "Calcium Induced Nonideal Mixing in Liquid-Crystalline Phosphatidylcholine-Phosphatidic Acid Bilayer Membranes." *Langmuir* 16(4): 1662-1667

REVIEWERS' COMMENTS:

Reviewer #1 (Remarks to the Author):

The authors fully addressed the issues raised in both reviews. In my opinion, the revised manuscript is significantly improved with regard to the original version. In particular, multiple minor but still important issues were clarified in the revised text, as suggested by the reviewers. Hence, I recommend this manuscript for publication in Nature Communications Chemistry.

Reviewer #2 (Remarks to the Author):

The authors have satisfactorily addressed all my concerns and I recommend the manuscript to be published in its current form.

I also like the new title.